# Structural basis of ion uptake in copper-transporting P$_{1B}$-type ATPases

Nina Salustros [1], Christina Grønberg[1], Nisansala S. Abeyrathna[2], Pin Lyu[1,3], Fredrik Orädd[4], Kaituo Wang [1], Magnus Andersson [4], Gabriele Meloni [2] & Pontus Gourdon [1,5] ✉

Copper is essential for living cells, yet toxic at elevated concentrations. Class 1B P-type (P$_{1B}$-) ATPases are present in all kingdoms of life, facilitating cellular export of transition metals including copper. P-type ATPases follow an alternating access mechanism, with inward-facing E1 and outward-facing E2 conformations. Nevertheless, no structural information on E1 states is available for P$_{1B}$-ATPases, hampering mechanistic understanding. Here, we present structures that reach 2.7 Å resolution of a copper-specific P$_{1B}$-ATPase in an E1 conformation, with complementing data and analyses. Our efforts reveal a domain arrangement that generates space for interaction with ion donating chaperones, and suggest a direct Cu$^+$ transfer to the transmembrane core. A methionine serves a key role by assisting the release of the chaperone-bound ion and forming a cargo entry site together with the cysteines of the CPC signature motif. Collectively, the findings provide insights into P$_{1B}$-mediated transport, likely applicable also to human P$_{1B}$-members.

Copper is a transition metal executing vital functions within cells and is therefore an essential micronutrient for organisms within all kingdoms of life. Its ability to redox-cycle between reduced (Cu$^+$) and oxidized (Cu$^{2+}$) states is exploited in a palette of key enzymes, e.g. the cytochrome c oxidase or NADH dehydrogenase, critical for fundamental metabolic processes such as cellular respiration[1]. Nevertheless, free intracellular copper can trigger the formation of toxic hydroxyl radicals and displace other metals from proteins[2]. Consequently, intracellular copper levels are tightly regulated, mediated by numerous dedicated chaperones, regulators and export proteins, most notably copper-transporting P-type ATPases.

P-type ATPases comprise a large superfamily that couple the energy from ATP hydrolysis to the transport of cargo across biological membranes. They are sub-divided into five subfamilies, P$_{1-5}$, with up to four subclasses, A-D, based on sequence identity and transport specificity[3], which ranges from metal cations to phospholipids[4], polyamines[5] and transmembrane helices[6]. The transition metal-specific subclass 1B (P$_{1B}$-ATPases) catalyzes efflux of e.g. Zn$^{2+}$, Co$^{2+}$, Fe$^{2+}$ from the

cell interior and includes copper-specific members that are historically sub-divided into P$_{1B-1}$- (CopA) and P$_{1B-3}$-ATPases (CopB)[7,8]. P$_{1B}$-ATPases are abundant in prokaryotes, protecting the organism from heavy metal stress. Two CopA proteins are present in humans, ATP7A and ATP7B. Malfunction of these members causes the severe neurological disorders Menkes and Wilson disease, respectively[9,10].

P-type ATPases share a common overall architecture consisting of three cytosolic (actuator (A-), phosphorylation (P-) and nucleotide binding (N-)) domains, together with a membrane-spanning (M-) domain composed of six omnipresent transmembrane helices, M1-M6 (Supplementary Fig. 1). In addition, P$_{1B}$-ATPases possess two N-terminal transmembrane helices MA and MB, and one to six typically N-terminal heavy metal binding domains (HMBDs), the latter having an enigmatic role for the function linked to ion-uptake and/or regulation[11–13]. The superfamily follows the so-called Post Albers reaction cycle, an alternating access mechanism defined by inward-facing E1 and outward-facing E2 conformations with high and low affinity, respectively, for the transported cargo from the cytoplasmic (in) to

[1]Department of Biomedical Sciences, Copenhagen University, Maersk Tower 7-9, Nørre Allé 14, DK-2200 Copenhagen, Denmark. [2]Department of Chemistry and Biochemistry, The University of Texas at Dallas, 800W Campbell Rd., Richardson, TX 75080, USA. [3]Department of Biology, University of Copenhagen, Universitetsparken 13, DK-2100 Copenhagen, Denmark. [4]Department of Chemistry, Umeå University, Linneaus Väg 10, SE-901 87 Umeå, Sweden. [5]Department of Experimental Medical Science, Lund University, Sölvegatan 19, SE-221 84 Lund, Sweden. ✉e-mail: pontus@sund.ku.dk

extracellular/luminal (out) sides (Fig. 1)[14,15]. Cargo uptake and occlusion from the cytosol is accomplished in E1 conformations, and together with ATP-dependent phosphorylation of an invariant, catalytic aspartate in the P-domain yielding the E1P form, this leads to the transition to the outward-facing E2P state. The cargo is then released to the extracellular space and upon dephosphorylation (E2), the pump returns to the inward-facing E1 conformation.

Thus far, structural information on the P$_{1B}$-ATPase core is limited to structures of CopA from *Legionella pneumophila* (LpCopA), ATP7B from *Xenopus tropicalis*, and ZntA from *Shigella sonnei* in E2P and E2.P$_i$ conformations[16–19], as well as individual soluble domains and a low-resolution cryo-EM model of CopA from *Archaeoglobus fulgidus* (AfCopA)[20–22]. Consequently, no metal-bound (E1) conformation of a P$_{1B}$-ATPase has been reported, leaving basic mechanistic principles regarding ion uptake, transfer and binding elusive. Typically, cuproproteins deliver copper to P$_{1B}$-ATPases in vivo, i.e., *Archaeoglobus fulgidus* CopZ to AfCopA[23] or homologous Atox1 to human ATP7A/B. However, although it has been proposed that an electropositive platform, MB′, in-between MB and M1 is involved in the recruitment of CopZ[24], the molecular determinants of chaperone-mediated Cu$^+$ transfer to the ATPase are elusive. Here, we report three inward-facing E1 structures of a copper-transporting P-type ATPase, providing fundamental insights into ion uptake and transport by the entire P$_{1B}$-subclass.

## Results and Discussion
### Structure determination
To shed further light on the ion entry and transport mechanism of P$_{1B}$-ATPases, we targeted CopA from the model organism *Archaeoglobus fulgidus* using a truncated construct lacking both the N- and C-terminal heavy metal binding domain (AfCopAΔNΔC). Crystallization was conducted in the presence of high concentrations of lipid and detergent (the HiLiDe method[25]) and 0.5 mM CuSO$_4$ which yielded crystals diffracting to 2.7 Å resolution. The structure was determined using molecular replacement as a phasing method. The final model yielded R/R$_{free}$ of 22.5/25.7 (Supplementary Table 1) and enabled modeling of side chains in the A-, P- and most parts of the M-domain, whereas somewhat lower local resolution was observed in the N-domain, likely due to the absence of nucleotide and the lack of crystal contacts in that region (Supplementary Figs. 2–3 & Methods).

### A unique E1 conformation
The recovered AfCopA structure exhibits the anticipated P$_{1B}$-ATPase architecture[16–19,26], including the cytosolic A-, P- and N-domains and a total of eight membrane-spanning helices, MA, MB and M1-M6, clearly separating these molecular pumps from other types of P-type ATPases (Fig. 1a & Supplementary Fig. 1). However, the nucleotide-binding pocket of the N-domain is facing the catalytic aspartate (D424) in the P-domain, while there is a large distance between the A- and N-domains, clearly deviating from previously determined conformations of P$_{1B}$-ATPases. To integrate the determined AfCopA structure into the overall P-type ATPase reaction cycle, we performed structure-based alignments of the soluble domains to the available structures of the well-studied Ca$^{2+}$-transporting sarco/endoplasmic reticulum P-type ATPase (SERCA), revealing highest similarities to [Ca]$_2$ E1·ATP conformations (PDB-IDs 3N8G, 1T5S & 1T5T) (Supplementary Fig. 4)[27]. This was unexpected, as the AfCopA crystals were generated in the presence of cargo, but without ATP or non-hydrolysable analogues thereof, i.e., equivalent to conditions that capture early E1 states of SERCA. In addition, crystals of AfCopA obtained in the absence of copper and nucleotide yielded a structure in a similar overall conformation, again indicative of an early E1 conformation and perhaps an E1 (ground state) preference for this particular CopA member (Supplementary Figs. 3–5 & Methods).

However, the positions of the conserved dephosphorylation ((T/S)GE) and nucleotide binding (HP) motifs of the A- and N-domains, respectively, relative to the P-domain in the determined AfCopA structure do not reveal similarities to any of the available structures of SERCA (Fig. 1 & Supplementary Figs. 4 & 6). This contrasts to the previously reported E2P and E2.P$_i$ structures of P$_{1B}$-ATPases, which superpose well with the corresponding SERCA conformations (Fig. 1b & Supplementary Fig. 4). The peculiar arrangement of the AfCopA soluble domains in the determined E1 state relative to SERCA becomes particularly clear by considering the position of the invariant (T/S)GE dephosphorylation loop of the A-domain. In the AfCopA structure, it is placed at the end of M4, while it is located on the other side of the P-domain in E1 structures of SERCA (PDB-IDs 4H1W & 3N8G)[28,29] (Fig. 1b & Supplementary Fig. 7).

Noteworthy, the configuration of the AfCopA soluble domains rather resembles the recently determined E1 and E1P states of the Class 1A P-type ATPase KdpB, which is part of the KdpFABC complex[30,31] (Supplementary Fig. 7). The observed similarities between P$_{1A}$- and P$_{1B}$-ATPases can be rationalized by the shared evolutionary origin and alike topology of the subfamilies, i.e., the absence of an N-terminal A-domain extension and helices M8-10, which are present in other classes of P-type ATPases. However, the integration of KdpB into the KdpFABC complex makes it an atypical member of the P-type ATPase superfamily, and P$_{1B}$-ATPases likely operate in a profoundly different manner. We therefore conclude that the determined AfCopA structure resembles a deviant E1 conformation, likely affecting the P$_{1B}$-ATPase mode of action.

### The A-domain modulates the catalytic cycle
Structural comparisons to the previously determined E2.P$_i$ structure of LpCopA[16] disclose major rearrangements during the E2 → E1 transition (Fig. 2 & Supplementary Fig. 8). Most noticeably, a large A-domain rotation of approximately 100° relative to the P-domain moves the conserved (T/S)GE dephosphorylation motif from the catalytic aspartate of the P-domain, positioning it immediately in the vicinity of the end of M4, where it is stabilized by electrostatic interactions with the P-domain (Supplementary Fig. 9). The CopA A-domain and (T/S)GE motif undergo a counterclockwise rotation relative to the P-domain as seen from the cytosol. Along with the displacement of the A-domain, the N-domain is tilted, thereby closing the cytosolic headpiece, priming the nucleotide-binding pocket for ATP binding (Supplementary Fig. 10).

The E2 → E1 counterclockwise rotation of the A-domain is similar to the situation in KdpB[31] but opposite to SERCA, where the TGE motif rotates clockwise (Supplementary Fig. 7). But what is the mechanistic consequence and origin of this fundamental difference between P$_1$- and P$_2$-ATPases? The distinct location of the A-domain in CopA and SERCA must be dictated by the overall architecture. As indicated above, P$_{1B}$-ATPases lack the N-terminal A-domain extension and, consequently, the A/M1-linker. In SERCA, this connecting stretch determines the interspace between M1 and the distal portion of the A-domain, leaving the distance essentially constant in-between the E2.P$_i$ and E1 states (Supplementary Fig. 11). This is of particular importance as the length of the A/M1-linker is known to be critical for the SERCA reaction cycle, with insertions or deletions abolishing ATPase activity[32]. Conversely, due to the absence of the N-terminal A-domain extension in P$_{1B}$-ATPases, the A-domain is less conformationally restricted, and the aforesaid distance decreases from 69 to 65 Å during the E2.P$_i$ → E1 transition (Supplementary Fig. 11).

Analogously, modifications of the A/M3 linker, which generally adopts an elongated loose shape in E1 states while it is more structured in E2 conformations, affects the function of SERCA[33,34]. This transport cycle shift in SERCA is likely supported by stabilizing electrostatic interactions between the A- and P-domains in the E2 conformations[33]. However, the parts of the P- and A-domains forming these interactions

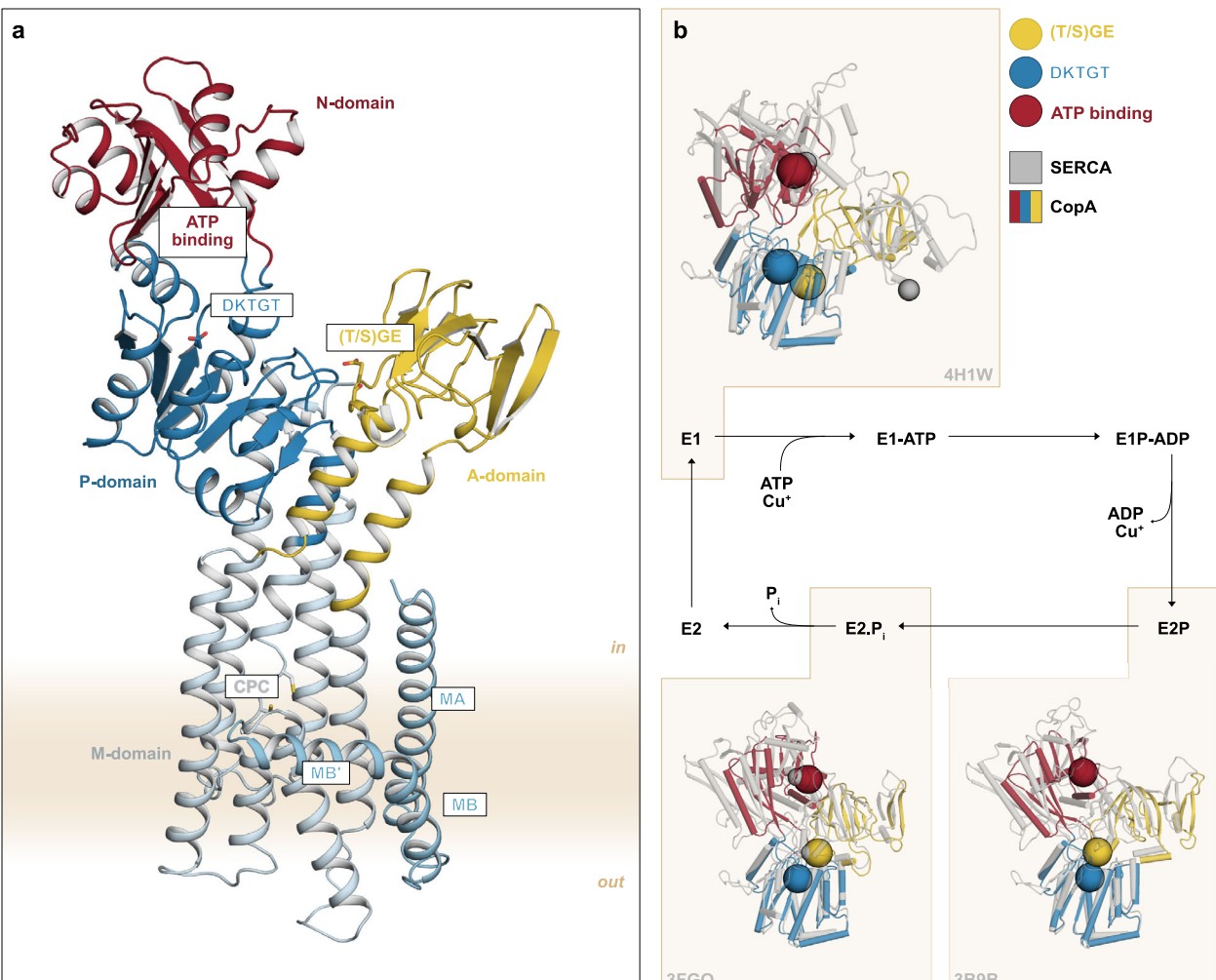

**Fig. 1 | Overall structure of the E1 conformation. a** The structure shows the typical P-type ATPase domain setup, with three cytosolic A- (colored in yellow), P- (blue) and N- (red) domains and a membrane-spanning M-domain. The P$_{1B}$-specific MA-MB helices are shown in cyan and M1-6 in grey. Functionally important features and motifs are highlighted. **b** Post-Albers reaction cycle. Insets demonstrate

P-domain structural alignments of the determined AfCopA (E1 state) and the available LpCopA (E2P and E2.P$_i$; PDB-IDs 4BBJ & 4BYG) structures (in colors) to the of corresponding states of SERCA (in grey; 4H1W, 3B9B & 3FGO), with the M-domains removed for clarity. The E2 states demonstrate considerable overlap with SERCA, while the determined E1 structure displays a distinct organization.

---

are absent in P$_{1B}$-ATPases (Supplementary Fig. 9). Lack of these features confers an overall electronegative surface patch to the P-domain of CopA, which permit interaction with electropositive residues of the A-domain also in the determined E1 conformation (Supplementary Fig. 9). As a consequence, the A/M3 linker is shortened by 4.6 Å with the E2.P$_i$ → E1 transition, while the distance is enlarged with 3.3 Å in SERCA (Supplementary Fig. 11).

Collectively, this demonstrates that the push-and-pull forces exerted on the A-domain during the course of the transport mechanism are significantly different in-between P$_1$- and classical P$_2$-ATPases, in agreement with the detected atypical arrangement of the A-domain of CopA. It is possible to speculate that the postulated absence of counterions and structural differences such as the missing A/M1 linker in P$_{1B}$-ATPases[11] leads to the need for alternative stimulation of the E2.P$_i$ → E1 transition. The tighter restrains of the A/M1 and A/M3 linkers in the E1 state may serve such as purpose, powering the A-domain rotation required for reaching the E1 conformation.

**Copper entry to the transport pathway - conserved residues in the M-domain become accessible from the cytoplasmic side**
Unexpectedly, in the M-domain, MA-M2 and M3-M6 each form a helix bundle, which move relative to one another in the E2.P$_i$ → E1 transition, as revealed by unbiased analyses of the relative locations of each

transmembrane helix in the two states (Fig. 2 & Supplementary Fig. 12). In the determined E1 structure, M2 and M6 converge in the intersection of these sub-domains. As a consequence, the previously proposed ion exit pathway becomes obstructed, rendering it inaccessible from the outside as apparent from analysis using the software HOLLOW[17,35] (Fig. 2a–d). In contrast, on the opposite side, the protein interior is exposed to the intracellular environment. Thus, the determined structure displays an inward-facing E1 conformation.

Notwithstanding the absence of structural information, copper binding to P$_{1B}$-ATPases has been extensively studied[36–38]. Based on the Pearson acid-base (HSAB) theory, it is anticipated that Cu$^+$ with a relatively small charge and "soft" nature will more favorably be coordinated by "softer" and more polarizable cysteines, methionines and to a lesser extent histidines in contrast to other metal ions[39]. Accordingly, ion uptake from the cytosol has been proposed to occur via an entry site formed by the conserved M158 (at the MB' - M1 transition), E205 (M2) and D336 (M3)[16,17,24]. An alternative sulfur-based model was introduced later, in which Cu$^+$ binds to a transient site formed by M158 together with C380 and C382 of the CPC motif located at a characteristic kink of M4[38]. The ion would then be shuttled from the entry site or/via the transient site to the high-affinity ligands in the M-domain. High-affinity ion binding is expected between transmembrane helices M4-M6, harboring several invariant amino acids,

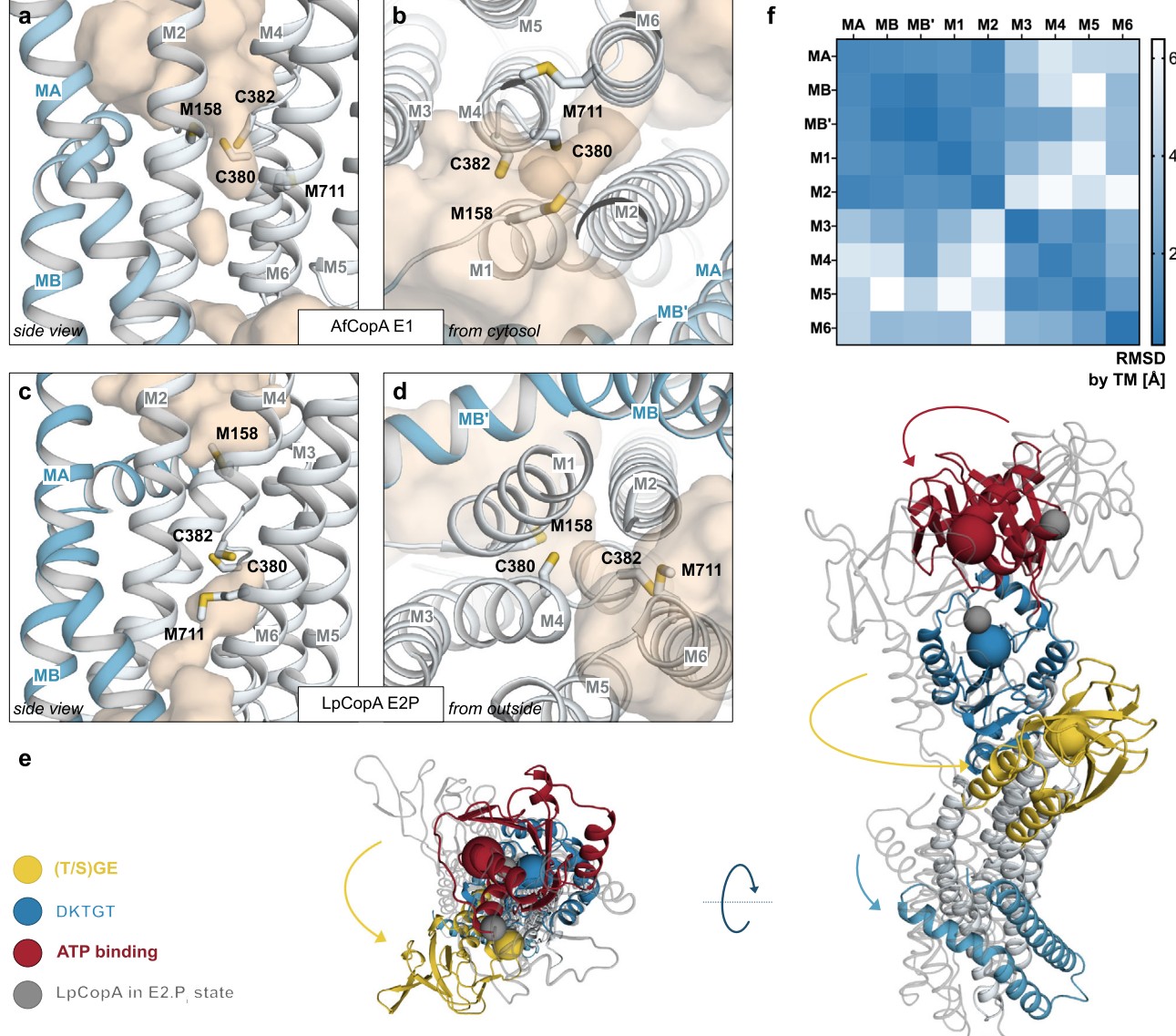

**Fig. 2 | The E2 → E1 transition.** M-domain solvent-accessible regions (wheat) of **a**, **b** the here determined AfCopA E1 structure and **c**, **d** the E2P state of LpCopA (PDB-ID 4BBJ with the residues numbered as AfCopA). In the AfCopA E1 structure, the conserved CPC motif in M4 is exposed to the cytosol, whereas the LpCopA E2 conformation is outward-open. **e** Alignment of the AfCopA E1 state to the P-domain of LpCopA (PDB-ID 3RFU) illustrates conformational changes during the E2.Pi → E1 transition, most strikingly a large rotation of the A-domain. **f** Distance difference matrix of the M-domain detecting relative movements between helices[63]. For this purpose, a homology model of AfCopA based on the E2.Pi structure of LpCopA was generated using SwissModel[52], and MA-M6 were assigned manually. MA-M2 and M3-M6 hardly rearrange relative to one another, hence forming two separate bundles of helices, whereas changes of up to 6 Å occur in between the sub-domains.

including the CPC residues as well as Y682, N683, M711, S714, and S715 of the YN (M5) and MxxSS (M6) motifs[8].

In the determined AfCopA structure, unlike the E2P and E2.Pi conformations of LpCopA[16,17], C380 and C382 are surface exposed, oriented towards M1-2 including M158 in the final model, again indicative of an early E1 conformation congruent with ion uptake (Figs. 2a & 3a). However, while the arrangement of M158, C380, and C382 is in agreement with Cu+ coordination in conjunction with the above-mentioned transient site, we were unable to unambiguously assign metal presence in the high-resolution data set (Fig. 4d). Conversely, Y682, N683, and M711 are facing the center of M4-6, primed for subsequent high-affinity Cu+ binding, hence including a substantial displacement of M711 relative to the E2.Pi conformation where it is exposed to the surrounding membrane (Fig. 3d). To investigate the functional significance of these conserved residues on AfCopA-mediated copper transport, we performed in vitro ATPase activity

measurements on protein reconstituted into liposomes (Methods). The determined Cu+-stimulated turn-over of wild type AfCopAΔNΔC was $242 \pm 26$ nmol Pi mg$^{-1}$ min$^{-1}$. Alanine substitutions of the M4-6 residues abolished the function while less dramatic effects were detected for the amino acids previously proposed to be involved in ion uptake (Fig. 3a), which is in agreement with earlier observations on CopA proteins[17,38,40]. In addition, we measured Cu+ binding to AfCopAΔNΔC and selected alanine substitutions by Inductively Coupled Plasma - Mass Spectrometry (ICP-MS), revealing an average stoichiometry of $0.87 \pm 0.10$ Cu+ ions per AfCopAΔNΔC protein (Fig. 3b). Cu+ binding was particularly impaired by mutation of the cysteine residues C380 and C382 in the conserved CPC motif in M4, but also the YN (M5) and MxxSS (M6) motifs (Fig. 3b). Therefore, our data support the requirement of numerous conserved residues in M4-6 for CopA-mediated Cu+ transport, while the orientation of these side chains in the determined structure suggests an early E1 conformation, as

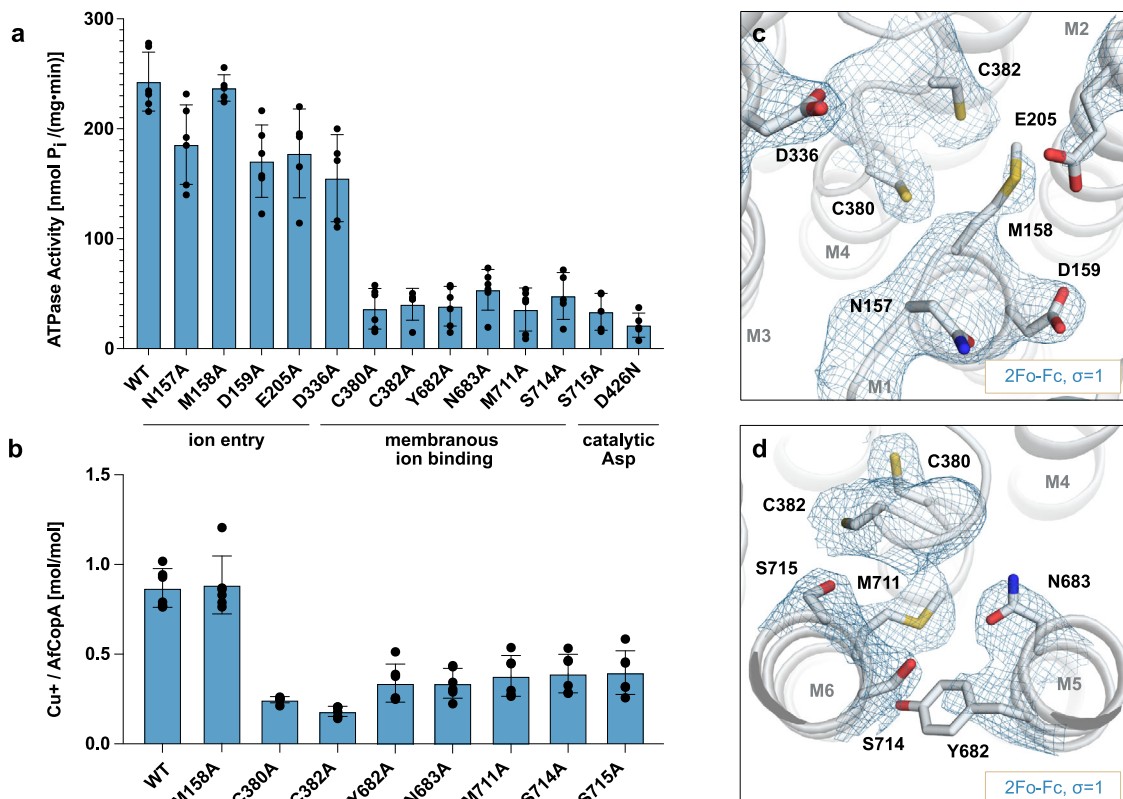

**Fig. 3 | Functional characterization. a** Cu⁺-stimulated ATPase activity of selected mutant forms as determined by a colorimetric in vitro ATPase assay based on protein reconstituted in liposomes. WT wild type AfCopAΔNΔC. Data were collected on material from two independent purifications with $n = 5$ (M158A, E205A, D336A, C382A, S714A, S715A, D426N) or $n = 6$ (other mutants), and are presented as mean ± SD. **b** Cu⁺ binding stoichiometries determined by ICP-MS measurements of selected mutant forms. Data were collected on material from two independent purifications with $n = 5$ (E205A) or $n = 6$ (other mutants), and are presented as mean ± SD. Source data are provided as a Source Data file. **c, d** Close-views of the AfCopA E1 structure as observed from the cytoplasmic side of the relevant residues in the entry region **c** and putative high-affinity ion binding region **d** with the final electron density at σ = 1 shown in blue mesh.

## Cu⁺ binding to, and conformational flexibility around, a transient entry site

The number and location of the high-affinity Cu⁺ binding sites in P₁ᵦ₋₁-ATPases is debated, representing a key unresolved issue in the field. Despite the open configuration of our structure and with the intention to further study the details of the physico-chemical environment of the transient site, we collected X-ray crystallographic data at the copper absorption edge (1.37 Å) diffracting to 3.3 Å. MR-SAD phasing in phenix.autosol[41] using the final AfCopA E1 structure as a search model resulted in a substructure with indications of a single Cu⁺ site close to C382 of the CPC motif, without sign of copper between M4-6, although it cannot be excluded that the signal relates to a non-copper atom (Fig. 3 & Supplementary Table 1). Further refinement yielded 60% occupancy of the Cu⁺ ion at the detected position, with maintained overall structure and with a similar arrangement of the region surrounding the Cu⁺ site compared to the data set collected at 1 Å wavelength. Specifically, both cysteines of the CPC motif are facing M158 interspaced by 4.4 and 5.1 Å, congruent with a trigonal-planar Cu⁺ coordination. Indeed, molecular dynamics (MD) simulations of the determined structure inserted in an in silico membrane, support coordination of Cu⁺ of M158, C380, and C382, following uptake from the surrounding aqueous environment (Supplementary Fig. 13a,b). However, alanine substitutions of M158 and neighboring N157 and D159 in the exploited liposome assay do not significantly alter Cu⁺-stimulated ATPase activity in vitro, in contrast to a previous report with alternative experimental settings (see

further below) (Figs. 2a & 3a)[42]. Furthermore, alanine substitution of M158 does not alter Cu⁺ binding stoichiometry to AfCopAΔNΔC as revealed by ICP-MS (Fig. 3b), indicating a role in Cu⁺ uptake rather than high-affinity Cu⁺ binding. In this context, it is worthwhile mentioning that the obtained electron density map appears poorly resolved around the CPC motif, including the side chain of the M158. We were therefore interested in further assessing the dynamics around the Cu⁺ entry site, and applied the ensemble refinement approach implemented in Phenix[43,44] (Fig. 4b & Supplementary Fig. 14). The obtained models are in good agreement with the overall data quality, i.e. with only subtle variations within the A-, P- and most parts of the M-domain, and with heterogeneity around the termini, periplasmic loops and the N-domain. However, as expected from the original data analysis, the CPC motif in the unwound part of M4, and the side chain of M158, feature conformational flexibility (Fig. 4b). Considering that the electron density maps is of high quality for all other parts of the M-domain, it is tempting to speculate that the detected dynamics reflect biological flexibility in the copper entry region in the determined E1 conformation. Along this line, the MD simulations exploring Cu⁺ entry resulted in interactions to M158, C380, C382, but not to the remaining conserved residues, which indicates that the protein conformation can coordinate the ion, but is not yet open to the internal high-affinity binding site (Supplementary Fig. 13). As such, the identified site resembles a flexible, transient Cu⁺ entry site for copper uptake, in agreement with the isolated early E1 conformation. Indeed, two complementing 100 ns MD simulations with a Cu⁺ initially placed in the trigonal sulfur position of AfCopA, indicate rapid transfer towards M711 together with two or three coordinating water molecules (Supplementary Fig. 13).

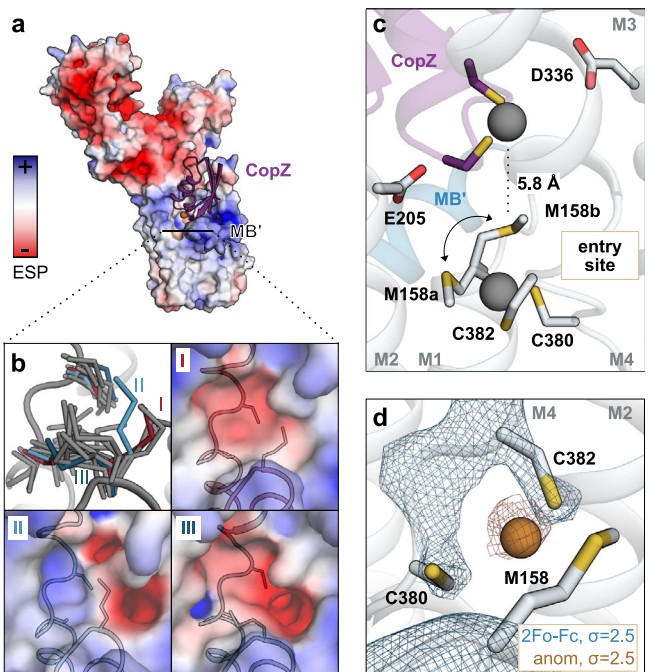

**Fig. 4 | Cu⁺ entry. a** Electrostatic surface representation of the determined CopA structure. CopZ docking was performed with pyDockWEB[45] using a homology model of the C-terminal part of AfCopZ (see Methods for details). **b** Ensemble refinement around the Cu⁺ entry site yielded multiple possible orientations of the M158 side chain. The electrostatic surface is shown for three selected ensembles (I-III), illustrating that modest reorientations have a major impact. **c** Model for Cu⁺ uptake based on CopZ docking and ensemble refinement. A transient Cu⁺ uptake site is likely formed by the Cu⁺-coordinating residues (the so-called CxxC motif) of CopZ and M158, adopting an upward-facing side chain conformation similar to ensemble II. Cu⁺ transfer from CopZ directly to M158 is possible, likely without involvement of E205 and D336 that were previously proposed to be involved in ion uptake[16,24]. Reorientation of M158 leads to formation of the transient Cu⁺ entry site detected in our data together with C380 and C382 of the CPC motif in M4. **d** Anomalous data collected at the Cu edge compatible with the presence of a Cu⁺ ion bound to C382 at the entry site. The final 2Fo-Fc and anomalous difference maps are shown in blue and brown mesh, respectively.

## CopZ-mediated Cu⁺ delivery

How is Cu⁺ uptake from the milieu then accomplished in P₁B-ATPases? Noticeably, the determined AfCopA structure features low electronegativity in the Cu⁺ uptake region surrounding M158, C380 and C382. Investigation of single ensembles reveals that only minor rearrangements around the Cu⁺ entry site strongly affect the electrostatic surface (Fig. 4a). In particular M158 has a strong effect on the charge exposed to the surroundings. However, with respect to the HSAB theory, such a negatively charged local environment may not be necessary for copper access and delivery. Instead, the presence of a coordination environment rich in sulfur-containing "soft" amino acid ligands (such as methionines and/or cysteines) as observed here may drive the Cu⁺ transfer event.

In this context, it is also of importance that in contrast to other ions, cytosolic Cu⁺ is at all times bound to chaperones, and according to the current model, expected to be transferred directly from these small soluble proteins such as CopZ to CopA[24].

To further investigate this delivery, we performed computational docking of an homology model of the C-terminal part of CopZ from *A. fulgidus* to the determined E1 conformation using pyDockWEB[45], suggesting that CopZ fits well in the groove between MB' and M2-4 (Supplementary Fig. 15). In this interaction model, the generally electronegative CopZ faces the electropositive MB' platform, supporting the notion that electrostatic complementation guides CopZ binding to

CopA[24]. Certain of the conformations of M158 detected by our ensemble refinement approach are positioned within 6 Å of the CopZ-bound Cu⁺-ion (Fig. 4c), illustrating that direct Cu⁺ transfer from CopZ to M158 may well be possible. Such a direct transfer of Cu⁺ would require the presence of yet another transition site, in which M158 is involved facing the cytosol, together with the CxxC cysteines of CopZ (Fig. 4). Cu⁺ would then be handed over to the entry site formed by M158, C380 and C382, and the membranous high-affinity binding site(s). We speculate that the surface exposed E205 and D336 are involved in Cu⁺ uptake by creating a suitable environment to stimulate release from CopZ, or through stabilization of the CopA - CopZ complex, unifying the available mechanisms of copper uptake.

Previous biochemical studies comparing the in vitro ATPase activity of wild type CopA stimulated by free (using cysteine as a vehicle) Cu⁺ or Cu⁺-loaded CopZ revealed that only the Cu⁺-CopZ-stimulated ATPase activity was abolished through alanine substitutions in the entry area, while free Cu⁺-stimulated ATPase activity remained unchanged[24]. This not only agrees with our data on AfCopAΔNΔC (Fig. 3a), but also emphasizes the specific role of M158, E205 and D336 in ion uptake. Since Cu⁺ is delivered to the ATPase core by chaperones, these residues in the entry area seem to be important in vivo, likely because they generate an appropriate docking and transfer region for Cu⁺ release from the chaperones combined with steric inability of the chaperones to directly reach the membranous high-affinity Cu⁺ binding ligands. Cysteine-bound Cu⁺ may well evade this requirement, replacing M158 as donors to the CPC cysteines. Thus, our E1 structure also provides a plausible Cu⁺ donation hypothesis to CopA proteins, combining the previous, somewhat conflicting models of ion uptake.

## Transport and regulation mechanism

Based on detailed analyses of the structurally determined E1 state and comparisons to already available structures combined with functional characterization, we propose that copper specific P₁B-ATPases operate using the following transport mechanism (Fig. 5). The E2.Pᵢ → E1 transition occurs along with major conformational changes, in particular a large rotation of the A-domain, likely triggered through phosphate release upon dephosphorylation in agreement with the conserved P-type ATPase alternating access mechanism. In conjunction, MA-M2 shift relative to M3-6, exposing the P₁B-type ATPase-specific CPC motif of M4 to the cytoplasm (Figs. 2a & 4).

The E1 structure represents a Cu⁺ uptake state capable of binding to cargo-bound CopZ, as confirmed by computational docking. Alternatively, Cu⁺ may be provided to the ATPase core by the N-terminal HMBD, which is structurally homologous to CopZ, or small molecular ligands such as glutathione. The unique position of the A-domain, that is different to P₂- and P₃-ATPases, is required to create sufficient space for CopZ docking to the ATPase core, as its position in SERCA would sterically hinder the interaction. Orientation of the M158 side chain towards the cytosol, as determined using the ensemble refinement approach, then allows trigonal-planar Cu⁺ coordination together with the cysteines of the conserved CxxC motif of CopZ, i.e. a transient site formed by mixed CopA-CopZ residues without direct influence of E205 and D336 (Fig. 4), in agreement with the sulfur-based transport pathway previously proposed for LpCopA[38]. However, it cannot be excluded that E205 and D336 directly participate in the ion transfer, too. In the determined E1 state, the side chain conformations of M158, C380 and C382 allow for trigonal-planar Cu⁺ coordination (Figs. 3b & 4). This is supported by anomalous signal at this site, as detected in the data set collected at the Cu⁺ edge. The metal is presumably shuttled to the identified site via M158 and is then passed on to the high-affinity ion binding site(s) in later E1 states, likely via C380 and C382. We note that M711 has adopted a different configuration in the E1 structure, pointing towards the center of M4-6, compatible with the formation of a single high-affinity site (Fig. 2a–d). Release to the extracellular space can later take place via the release

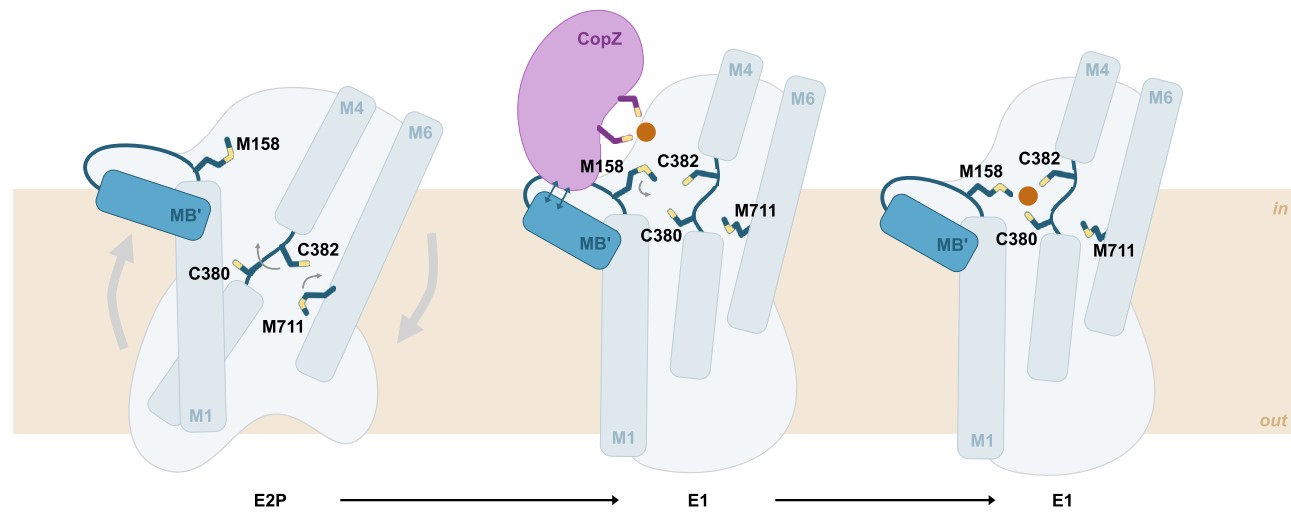

**Fig. 5 | Proposed ion uptake mechanism copper transporting P-type ATPases.** In the E2P state, the conserved CPC motif in M4 is exposed to the extracellular space. The transition to the inward-open E1 conformation occurs through movements of MA-M2 relative to M3-M6. Guided by electrostatic interactions, the cytosolic chaperone CopZ docks to the MB' platform. Cu⁺ is then handed over to the transient ion entry site in the M-domain, formed by a methionine (M158) at the cytosolic end of M1, and the two cysteines (C380 and C382) of the CPC motif in M4. Next, the metal is transferred to the high-affinity ion binding site(s), likely formed by the CPC motif and a methionine (M711) in M6, prior to the release to the extracellular space.

pathway detected in-between MA, M2 and M6[17], again in agreement with the MA-M2 and M3-6 subdomains revealed in this work. Regulation of $P_{IB}$-transport has previously been proposed to be achieved via the HMBDs, which have been shown to interfere with turn-over in copper-deficient conditions[13]. Interestingly, the location of the last two HMBDs in the *X. tropicalis* ATP7B E2.$P_i$ structure is structurally incompatible with inhibition in our E1 structure due to the arrangement of the A-domain[19] (Supplementary Fig. 16). Hence, it is possible that inhibition occurs in E2 states, with one HMBD binding in-between the A- and the P-domains, and that release of the HMBD then activates the transport.

In conclusion, we here present the structure of the model protein AfCopA, a copper-transporting $P_{IB}$-ATPase from *A. fulgidus*, in an inward-facing E1 conformation. In this state, the A-domain arrangement differs compared to other P-type ATPase subfamilies, suggesting the overall P-type ATPase transport mechanism is less conserved than previously assumed. Our data reveal the mechanism of Cu⁺ uptake from cytosolic chaperones and delivery to the ATPase core, in which M158 plays a pivotal role. These represent key findings that likely are transferable also to the human members, ATP7A and ATP7B. However, additional metal-bound structures of later E1 states are required to decipher the architecture of the high-affinity ion binding site(s) in $P_{IB}$-ATPases.

## Methods
### Large-scale expression and purification of AfCopAΔNΔC
AfCopAΔNΔC (residues G80-G736) (UniProtKB - O29777) was cloned into pET-22b(+) and expressed in *Escherichia coli* (C43 strain). The cells were cultured in LB medium supplemented with 0.1 mg/mL ampicillin at 100 rpm, 37 °C in baffled flasks until the optical density (OD600) reached 0.8 – 1. Next, protein expression was induced through the addition of 1 mM isopropyl β-D-1-thiogalactopyranoside (IPTG) and the temperature was lowered to 20 °C. Following 16 h the cells were harvested through centrifugation at 5000 × g for 15 min. Cells were resuspended in buffer A (20 mM Tris-HCl pH = 7.6, 200 mM KCl, 20% (v/v) glycerol) at 5 mL per g of cells and stored at −80 °C until further use. The cell suspension was supplemented with 5 mM β-mercaptoethanol (BME), 4 µg/mL DNase I, 1 mM MgCl₂, 1 mM phenylmethylsulfonyl fluoride (PMSF) and Sigma protease inhibitors (1 tablet per 6 L cell culture) prior to cell disruption at 25 kpsi in a high-pressure homogenizer (Constant System). Cellular debris was

removed via 30 min centrifugation at 20,000 × g, followed by the collection of membranes by 3 h centrifugation at 185,500 × g. Membranes were resuspended in buffer B (20 mM Tris-HCl pH = 7.6, 200 mM KCl, 20% (v/v) glycerol, 1 mM MgCl₂, 5 mM BME) at 10 mL per g of membranes. Membranes were stored at −80 °C until further usage. Membranes were solubilized in 1% (w/v) n-dodecyl β-D-maltopyranoside (DDM) at 4 °C for 2 h at 3 mg/mL protein concentration, as determined by the Bradford Assay. Insolubilized membranes were pelleted by 1 h ultracentrifugation at 185,500 × g. 30 mM imidazole and 500 mM KCl was added to the supernatant, and the sample was then loaded on a 5 mL HiTrap Chelating HP column (Cytiva). The column was washed with 10 column volumes (CV) of buffer C (20 mM Tris-HCl pH = 7.6, 200 mM KCl, 20% (v/v) glycerol, 1 mM MgCl₂, 5 mM BME, 0.15 mg/mL octaethylene glycol monododecyl ether ($C_{12}E_8$, from Nikkol)) containing 50 mM imidazole, and subsequently eluted in buffer C with increasing imidazole concentration, ending with 5 CV of 500 mM imidazole. Fractions containing AfCopAΔNΔC were pooled, and TEV protease was added in 1:10 (w/w) ratio. The sample was dialyzed against 20 mM Tris-HCl pH = 7.6, 80 mM KCl, 20% (v/v) glycerol, 1 mM MgCl₂, 5 mM BME, 0.15 mg/mL $C_{12}E_8$ at 4 °C for 16 h to remove imidazole. The following day, the sample was supplemented with 30 mM imidazole, and solid KCl was added to 500 mM final concentration. The sample was again loaded on a 5 mL HiTrap Chelating HP column (Cytiva), the flow-through containing cleaved AfCopAΔNΔC was collected, assessed using SDS-PAGE, and concentrated to approximately 20 mg/mL in a Vivaspin concentrator (MWCO = 50 kDa). At this stage, 8–10 mg of purified AfCopAΔNΔC were routinely obtained from 1 L of cell culture. 50 mg protein were centrifuged at 20.000 × g for 10 min prior to injection on a 120 mL self-packed Superose™ 6 (Cytiva) prep grade column. The protein was eluted in buffer D (20 mM Tris-HCl pH=7.6, 80 mM KCl, 20% (v/v) glycerol, 3 mM MgCl₂, 5 mM BME, 0.15 mg/mL $C_{12}E_8$), and fractions containing AfCopAΔNΔC were pooled and concentrated to 10 mg/mL in a Vivaspin concentrator (MWCO = 50 kDa). The protein was flash-frozen in liquid nitrogen and stored in aliquots at −80 °C until further usage. Purification results are shown in Supplementary Fig. 17.

### Crystallization
Macromolecular crystallization trials were conducted using the HiLiDe (high concentrations of lipid and detergent) approach[25]. A 25 mg/mL

stock solution of 1,2-Dioleoyl-sn-Glycero-3-Phosphocholine (DOPC) (Anatrace) in chloroform was prepared, and a thin lipid film was generated in a glass vial through chloroform evaporation under a nitrogen stream. 10 mg/mL AfCopA protein sample and $C_{12}E_8$ was added to obtain final concentrations of 2.25 mg/mL DOPC and 5 mg/mL $C_{12}E_8$. The sample was incubated for 16 h at 4 °C with gentle stirring. Aggregates were removed through ultracentrifugation at 50,000 × g for 20 min, after which 1 mM $CuSO_4$ was added to the supernatant (no supplementation was made for the apo crystals). Crystals were grown using the hanging drop vapor diffusion method, in 24 well plates containing 400 μL reservoir solution supplemented with 5 mM BME, in drops made of 1 μL protein sample and 1 μL reservoir solution. Crystals diffracting to 3.3 – 3.5 Å were obtained following 10 days in 1.5 M ammonium citrate, 0.1 M MES pH = 5.5. Final data diffracting to 2.8 Å were collected on a crystal for which 5.0 mM 06:0 Lyso PC (Avanti) was used as an additive. All crystals were fished using Litho-Loops (Molecular Dimensions) and flash-frozen in liquid nitrogen. Data were collected at Swiss Light Source, the Paul Scherrer Institute, Villigen, Switzerland, beam line X06SA.

### Structure determination and analysis

Collected data were processed and scaled using XDS[46]. Downstream structure determination and refinement was accomplished using Phenix[44]. Initial phases were obtained in PHASER[47] using molecular replacement, applying the structures of the soluble A- and PN-domains of AfCopA (PDB-IDs 2B8E & 3A1C)[20,21] together with the M-domain of LpCopA (PDB-ID 3RFU)[16] as search models. Initial model building was accomplished in Coot[48], and Rosetta refinement[49] was employed in the early stages of model building. The data enabled modeling of side chains in the A-, P- and M-domains, and side chain conformations in the N-domain were guided using available crystal structures (PDB-ID 3A1C)[21]. The final model includes residues 82-736, only lacking 4 residues at the N-terminus of the here exploited truncated form. All structural figures were generated using PyMOL[50], using the PyMol APBS electrostatics plugin[51] for electrostatic surface representations. To establish the AfCopA-AfCopZ docking model, a homology model of the CxxC-harboring C-terminal part of AfCopZ was generated in SwissModel[52] using the *E. hirae* CopZ structure (PDB-ID 1CPZ)[53] as a template. Docking to the determined AfCopA E1 structure (high-resolution data set) was performed using pyDockWEB with inputs N157-D159 (AfCopA), C149 + C152 (AfCopZ).

### Mutagenesis and point mutant production

Mutant plasmids were generated using the QuikChange Lightning Site-Directed Mutagenesis Kit (Agilent Technologies). Point mutants were expressed as the wild type in duplicates of 2 L LB medium per mutant, based on separate transformation colonies. Membrane preparation and purification was performed as described above for the wild type AfCopAΔNΔC, but with the following differences: Membranes from 2 L cell culture were solubilized for 2 h in 60 mL buffer B supplemented with 1% DDM. Solubilized material was incubated 1 h at 4 °C with 1 mL loose Ni Sepharose™ resin (Cytiva) equilibrated in buffer C, following which affinity purification was performed using gravity columns. The sample was loaded twice, and the resin was then washed first with 5 mL buffer C, followed by 10 mL buffer C with 50 mM imidazole. The protein was eluted in 5 mL buffer C with 500 mM imidazole, and 1 mg TEV protease was added per sample. On the following day, the sample was incubated with 1 mL fresh Ni resin equilibrated in buffer C for 1 h at 4 °C, and then loaded twice on gravity columns. The flow-through containing purified AfCopAΔNΔC mutants was collected and concentrated to 20 mg/mL. All the material (3-10 mg protein depending on the mutation) was injected to a Superose 6 Increase 10/300 GL column (Cytiva) and eluted in buffer D. Mutants were concentrated to >5 mg/mL, flash-frozen in liquid nitrogen and stored at −80 °C until further usage.

### Liposome preparation

For liposome preparation, a thin film of 20 mg DOPC was generated through chloroform evaporation of a stock solution under nitrogen stream and placed in a vacuum desiccator for 16 h. All buffers were treated with Chelex® 100 Chelating Resin (Bio-Rad) before usage. The lipid film was rehydrated in 20 mM MOPS-KOH pH = 6.8, 100 mM NaCl, 1 mM freshly prepared ascorbic acid, 10 mM freshly prepared cysteine, and placed for 3 × 15 min in a bath sonicator. The sample was exposed to 3 freeze-thaw cycles in liquid nitrogen, and subsequently extruded 11 times through 100 nm filters (Avanti). Liposomes were flash-frozen in liquid nitrogen and stored at −80 °C until use.

### In vitro ATPase assay

Liposomes were thawed and destabilized through addition of 0.01% (w/v) DDM for 1 h at 18 °C. Reconstitution was performed at a lipid:protein molar ratio of 20:1 with a final protein concentration of 0.5 mg/mL. Concentrated protein sample and liposomes were mixed with reconstitution buffer (20 mM MOPS-KOH pH = 6.8, 100 mM NaCl, 1 mM ascorbic acid) supplemented with 0.15 mg/mL $C_{12}E_8$ and incubated for 1 h at 4 °C. Detergent was removed through addition of Bio-Beads™ SM-2 resin following 1 h, 2 h and 18 h. Proteoliposomes were collected by centrifugation at 60.000 × g for 1.5 h, and were resuspended in reconstitution buffer. The samples were then used for measuring in vitro ATPase activity by employing an assay originally developed by Baginski and colleagues[54]. Briefly, 0.2 mg/mL proteoliposomes were added to 20 mM MOPS-KOH pH = 6.8, 100 mM NaCl, 1 mM ascorbic acid, 10 mM cysteine and 100 μM $CuCl_2$. The assay was performed at 65 °C and the samples were incubated 10 min before the reaction was started by adding 5 mM ATP (Sigma Aldrich). 50 μL of the respective reaction mixtures were transferred to a 96-well microplate and mixed with an equal volume of ascorbic acid solution (formed by mixing a solution containing 0.17 M ascorbic acid and 0.1 % SDS in 0.5 M HCl with aqueous 28.3 mM ammonium heptamolybdate in a 5:1 ratio, all from Sigma Aldrich) after 5, 12 and 18 min. Color development was stopped through addition of 75 μL of sodium arsenic solution (consisting of 0.068 M trisodium citrate, 0.154 M sodium metaarsenic and 2% v/v glacial acetic acid, all from Sigma Aldrich) following 10 min incubation at 18 °C. After 30 min incubation, the absorbance was measured at 860 nm. The presented data are from at least 5 independent measurements on samples from biological duplicates. As also noted by others[12,55], CopA samples exhibit high background activity in the absence of added copper, which we assume arises from trace levels of residual copper in our samples. Background activity of 30% was subtracted from plotted values (Supplementary Fig. 17). Figures representing ATPase activity measurements were generated using GraphPad Prism.

### ICP-MS measurements

$Cu^+$ binding stoichiometry in wild type AfCopA and mutants was determined by ICP-MS analysis. The purified protein stocks were diluted to final 1 mg/mL using a buffer containing 20 mM Tris pH = 7.6, 80 mM KCl, 20% (v/v) glycerol, 3 mM $MgCl_2$, 5 mM BME, 0.15 mg/mL $C_{12}E_8$. 4 $Cu^{2+}$ equivalents (from a $CuCl_2$ stock in $H_2O$) were added to the protein solutions and incubated at room temperature for 15 min. Excess and unbound copper was removed by injecting the protein solutions through a 5 mL HiTrap desalting column, pre-equilibrated with buffer. The concentration of eluted proteins was determined via a Bradford assay with Bovine Serum Albumin (BSA) as a standard. The eluted protein samples were digested in 8% (w/v) $HNO_3$ (Sigma-Aldrich) at 80 °C for 12 h. The samples were subsequently diluted to adjust the $HNO_3$ concentration to final 3% (w/v), and the copper content in samples were analyzed with an Agilent 7900 ICP mass spectrometer connected to a CETACASX-500 auto-sampler for sample injection. Control experiments were conducted for background correction

using protein buffer excluding protein. Figures representing ICP-MS measurements were generated using GraphPad Prism.

## Molecular dynamics (MD) simulations

Two ensembles that differed in the CPC region (denoted as I and II in Fig. 4b) were inserted into DOPC (1,2-dioleoyl-sn-glycero-3-phosphocholine) membrane patches using the CHARMM-GUI membrane builder[56]. Amino acid residue E457 was protonated as predicted by PROPKA3.1[57,58]. The systems were solvated in TIP3P water and 150 mM NaCl and energy minimized with a 5,000 step steepest-descent algorithm. The systems were equilibrated with gradual release of position restraints from the water and lipids for a total of 30 ns followed by 250 ns non-restrained production runs. Each protein conformation was also simulated in independent repeat simulations starting from a different set of initial velocities, adding up to a sampling total of 250 ns × 4. A Nose-Hoover temperature coupling[59] was applied using a reference temperature of 303.15 K. A semi-isotropic Parrinello-Rahman pressure coupling[60] was applied with a reference pressure of 1 bar and compressibility of 4.5e-5 bar$^{-1}$. The systems were simulated using the GROMACS-2021 simulation package[61] and CHARMM36 all-atom force fields[62]. The membrane domain was used as alignment reference for the root means square deviation, which showed a stable evolution for the protein backbone between 150-250 ns of each trajectory. We extracted one representative average structure from the 150–250 ns slice of all four trajectories and introduced a Cu$^+$ ion at a cytoplasmic position 5 Å above the pathway indicated by the CopZ docking analysis. In a short 20 ps simulation, the Cu$^+$ ion was then pulled in the z-direction along the membrane vertical at 1 Å/ps with a force constant of 1000 kJ mol$^{-1}$ nm$^{-2}$. To explore the interactions to potential coordinating residues, we also simulated the AfCopA structure with the Cu$^+$ ion initially placed in the position indicated from the experimental observations (as in Fig. 4d). Here, two independent systems were energy minimized and equilibrated for 2 ns with gradual release of position restraints on the protein, lipids and ions followed by 100 ns production simulations.

## Reporting summary

Further information on research design is available in the Nature Research Reporting Summary linked to this article.

# Data availability

The structural coordinates generated in this study have been deposited in the Protein Data Bank under accession codes: 7ROI (E1 in the presence of copper), 7ROH (E1 in the presence of copper with data collected at the copper edge), 7ROG (E1 in the absence of copper). Source data are provided with this paper. All data and materials supporting the findings in the manuscript are available from the corresponding author upon reasonable request. Source data are provided with this paper.

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

## Acknowledgements

The PhD studies of NS were funded by the Lundbeck Foundation. The post-doc fellowship of CG was supported by The BRIDGE - Translational Excellence Programme at University of Copenhagen funded by the Novo Nordisk Foundation. CG was also financially assisted by The memorial foundation of manufacturer Vilhelm Pedersen and wife - and the Aarhus Wilson consortium. PG is supported by the following Foundations: Lundbeck (R313-2019-774, R133-A12689 and R346-2020-2019), Knut and Alice Wallenberg (2020.0194 and 2015.0131), Carlsberg (CF15-0542), Novo-Nordisk (NNF13OC0007471), Brødrene Hartmann (A29519), Agnes og Poul Friis, Augustinus (16-1992), Craford (20200739, 20180652 and 20170818) as well as The Per-Eric and Ulla Schyberg (38267). Funding was also obtained from The Independent Research Fund Denmark (9039-00273 and 6108-00479), the Swedish Research Council (2016-04474 and 521-2012-2243) and through a Michaelsen scholarship. GM was supported by the National Institute of General Medical Sciences of the National Institutes of Health (R35GM128704), the Robert A. Welch Foundation (AT-1935-20170325 and AT-2073-20210327), and the National Science

Foundation (CHE-2045984). MA is supported by the Swedish Research Council (2020-03840). We acknowledge Jose M. Argüello for providing the AfCopA gene, and Julie Winkel Missel for help with liposome preparation. We are grateful for assistance with crystal screening and data collection at the Swiss Light Source, the Paul Scherrer Institute, Villigen, beam line X06SA. Access to synchrotron sources was supported by the Danscatt program of the Danish Council of Independent Research. The computations were performed on resources provided by the Swedish National Infrastructure for Computing (SNIC) through the High-Performance Computing Center North (HPC2N) and National Supercomputer Centre (NSC) under projects SNIC 2021/5-301, SNIC 2021/5-362, SNIC 2022/22-441, and SNIC 2022/5-168.

## Author contributions

C.G. and P.G. initiated the project. N.S., C.G. and P.G. contributed to identification of scientific problem, experimental planning, data analysis, interpretation. N.S. and C.G. performed cloning, overproduction and purification for the structural and functional studies, as also assisted by P.L. N.S. accomplished the crystals, NS and CG collected diffraction data, and determined the structure. K.W. provided support for the structure determination. N.S. and C.G. refined the structures. N.S. executed ATPase activity assays. N.A. performed ICP-MS measurements supervised by G.M. F.O. and M.A. performed M.D. simulations. N.S. prepared the figures. N.S., C.G. and P.G. wrote the first draft. All authors commented on the manuscript. C.G. and P.G. supervised the project.

## Funding

## Competing interests
The authors declare no competing interests.
