## [Peer Review File · Nature Communications]

Structural basis of ion uptake in copper-transporting P1B-type ATPasesREVIEWER COMMENTS

Reviewer #1 (Remarks to the Author):

The study reports an interesting and significant new step in determining the structural basis of metal transport by P1B-type ATPases. The data are clear and well-illustrated; the Supplement is very helpful. The only concern that I have is that the manuscript is somewhat difficult to read, especially for those outside the P-type ATPase field. The constant references to SERCA makes it hard to get a clear picture of how the AfCopA looks on its own and how it may work. Specifically:

1) Section "A unique E1 conformation"

a) References to previous SERCA and P1B-ATPase structure in the same paragraph are confusing. It could be more helpful to the reader if the authors first show that the current conformation of AfCopA is new/different compared to the previously published P1B-ATPase structures and after that compare it to known E1 conformations of other subfamilies to establish its uniqueness.

b) "...unique mechanistic features of P1A- and P1B-ATPases". While it is fair to point to some structural similarities between the two subfamilies of pumps, the overall structural and functional differences between them are too large and do not support the authors suggestions about their "unique mechanistic features". Please delete or re-phrase

2) Section: "The A-domain modulates the catalytic cycle".

a) Again, everything is intermixed. The authors may like to consider the following organization of the paragraph: "Structural comparisons to the previously determined E2.Pi structure of LpCopA disclose major rearrangements during the E2 → E1 transition (Fig. 2). Most noticeably, a large A-domain rotation of approximately 100° relative to the P-domain moves the conserved (T/S)GE dephosphorylation motif from the catalytic aspartate of the P-domain, positioning it immediately in the vicinity of the end of M4, where it is stabilized by electrostatic interactions with the P-domain (Supplementary Fig. 8). The displacement of the A-domain enables a tilt of the N-domain, closing the cytosolic headpiece, priming the nucleotide binding pocket for ATP binding (Supplementary Fig. 9)" After that one can talk about similarities and differences to other pumps.

b) "... . In agreement with this notion, the arrangement of the soluble domains has highest similarity". This phrase is confusing, since the authors stated in the previous section that this new conformation is unique. Please either clarify, reconcile with the above section, or delete

c) In SERCA, this connecting stretch orchestrates the interspace.. - What does it mean "orchestrates"? - occupies? controls the volume of the space? Please clarify

d) " It is possible to speculate that the postulated absence of counterions in P1B-ATPases leads to the need for alternative stimulation of the E2.Pi \rightarrow E1 transition, and the tighter restrains of the A/M1 and A/M3 linkers in the E1 state may

serve such as purpose, powering the A-domain rotation required for reaching the E1 conformation" - Protons, while not transported, are required for copper transport by the vertebrate P1B pumps. Does the structure of a recently published frog P1B-ATPase have similarly tight A/M3 restrain? If it does - then perhaps the noted structural differences between P1B and P2-type pumps may explain very different turnover rates, rather than differently used counter ion. Please consider and discuss

3) Section "Copper uptake - conserved residues in the M-domain become accessible from the cytoplasmic side" -

a)"Uptake" is a very specific term related to transport, typically, from the outside of the cells into the cytosol. It could be confusing to researchers outside of the P-type ATPase field. Please consider replacing with "Copper binding to transport site" o "Copper entry to the transport pathway"

b) "Unexpectedly, in the M-domain, MA-M2 and M3-M6 each form a.." - delete "Unexpectedly"

c) "Strikingly, in our structure, unlike the E2P and E2.P" - delete "strikingly"

4. Section "CopZ-mediated Cu⁺ delivery"

a) in the sentence "Noticeably, the determined..." please delete the " opposite to available E1 structures of SERCA and the Na⁺,K⁺, -ATPase 25,43 (Supplementary Fig. 14)" Given very different nature of transported ions this particular comparison to P2-pumps is somewhat meaningless and only distract the reader from the description of the Cu-entry site

b) The section on CopZ-mediated transfer is highly speculative and should be significantly toned-down or deleted. There are several problems with this part. While it is convenient to dock the C-terminal part of CopZ, in cells the Cu transfer is mediated by the entire protein and therefore the entire CopZ has to be considered. More importantly, in order for Cu transfer to occur the coordinating ligands must be much closer - i.e. less than 3Å, and it is not clear that even at this distance a single Met158 would be sufficient to facilitate the transfer of Cu from the CxxC based site of the chaperone.

Is there any example from the previous studies that "a suitable electronegative environment" on the acceptor protein stimulates release of copper from the chaperone? If such example exists, the authors should provide a reference.

Lastly, if E205 and D336 are involved in stabilizing interactions between the chaperone and the pump, one would expect that these residue would be less exposed and the interaction site will be altered in E2/E2P conformations, when chaperone presumably does not bind/deliver Cu to the transport site. Is this the case?

Suppl. Figure 12 - "Indeed, in molecular dynamics (MD) simulations of the determined structure inserted in an in-silico

membrane, M158, Cys380, and Cys382 coordinate a Cu⁺, following uptake from the surrounding aqueous environment" - Based on this simulation it is not clear whether Cys380 comes close enough to copper to provide coordination environment. Please clarify.

Reviewer #2 (Remarks to the Author):

Copper is essential to cells as a cofactor for many key enzymes, while excess copper is cytotoxic. Therefore, tight control of intracellular copper levels is crucial for cell. Copper-transporting Class IB P-type ATPases (CopA) are essential in this process by extruding Cu from the cytoplasm of cells. Like other P-type ATPases, CopA is proposed to transport substrate through a cyclic transition of E1-E1P-E2P-E2 states (Post-Albers model), with E1 and E2 states associated substrate with high and low affinity, respectively. By now, structural information on CopA is limited to structures of CopA from *Legionella pneumophila* (LpCopA) in E2P and E2.Pi states and frog ATP7B in E2.Pi state (a very recent published work: *Sci Adv.* 2022 Mar 4; 8(9):eabl5508). Because of lacking structures in other states, molecular mechanism of CopA remains unclear.

In this study, the authors report a cryo-EM structure of CopA in E1 conformation, and found dramatic domain arrangement in the structure. It's very exciting to see the central Cu binding site switch from facing out in previous E2pi state to facing cytosol in this structure. Furthermore, authors proposed the conformational change generate enough space for interaction with ion donating chaperones. Together with the functional results, the Cu transfer mechanism from the chaperone to the transmembrane core suggested by authors is with novel insights and pretty convincing.

Overall, this work is an important study, the paper is well written and the majority of the conclusions are supported by the data.

Several major issues were listed as below:

1. Considering the membrane embedded TMs is relative steady (only move in the membrane) during the substrate transport cycle, it would be helpful for readers to understand the conformational change of CopA in E1 and E2Pi by aligning their TMs. Actually, as shown in supplementary figure S11, the whole TM domain of CopA in E1 and E2 states are superimposable well, despite of some relative shifts between TMA-TM2 and TM3-6. Such alignment by TMs will present the relative movement of A/P/N domains, while fig.2e present the relative shift of A by P. In this way, comparison between Fig.2a and 2c could be improved by aligning either MA-M2 or M3-M6.

2. Authors proposed a CopA-CopZ binding model using pyDockWEB prediction software, and suggest CopZ fits well in the groove between MB' and M2-4 of CopA. I wonder whether it's possible to confirm their direct binding by in vitro interaction assays, such as pull-down, SPR? Furthermore, it's also helpful to know whether the CopA still keep the CopZ binding platform in E2Pi state?

Minor points

1. In the first line of Page 7, there is probably a duplicate word "site";
2. TMs and key residues in Fig.2a are not labeled;
3. It would be helpful to cite the new paper of ATP7B structure in Sci Adv.
4. In Fig.2, please add the rotation markers to show the view from 2a to 2b, 2c to 2d.

Reviewer #3 (Remarks to the Author):

The manuscript from Salustros et al. presents novel high-resolution structures of the cytoplasmic-facing E1 conformation of a copper-specific P-type ATPase. The E1 state for this P-type ATPases has not previously been characterized structurally. The crystallization construct excluded the heavy metal binding domains. In addition to the structural characterization, the authors also performed functional experiments to identify residues involved in Cu⁺ binding, allowing them to propose that a number of conserved residues are involved in the Cu⁺ transport, including a pair of cysteines. Using homology modelling and protein-protein docking, the authors furthermore illustrate a potential model for delivery of Cu⁺ to the transporter from an intracellular chaperone.

The manuscript is well-written and easy to follow and the accompanying figures are of high quality, and the data and the conclusions are compelling. I recommend publication with minor revisions.

I only have a couple of minor questions for clarification:

- 1) The MD simulations presented in Suppl. Fig. 12: As I understand the methods, the Cu⁺ ion was pulled from the intracellular side and into the transmembrane domain. Does the pulling stop at the suggested Cu⁺ site? If not, what happens at around 15 ps where the Cu⁺ seems to unbind? Is the Cu⁺ stable in the proposed site in (longer) unbiased simulations?

2) Fig 4, legend: “CopZ docking was performed with pyDockWEB using a homology model of the C-terminal part of AfCopZ (see Methods for details)” – I don’t think these details are found in the method section?

Reviewer #4 (Remarks to the Author):

This research group have been pioneers in advancing our understanding of the structure-function of P1B-type metal-transporting ATPases. The actions of these proteins have wide significance for example in nutritional immunity, inherited disease, the bio-recovery of at-risk metals and enzyme metalation in industrial biotechnology. The structure of a family member in the E1 state was previously missing and this manuscript reveals unanticipated features pertinent to metal loading from donors and potentially metal-dependent regulation via rotations of the A- and N-domains associated with ATP-binding.

Some changes are needed to the text: The closing paragraph of the introduction states that bacterial CopA proteins are 'dependent' on copper donating CopZ chaperones, but this is not as suggested. Some bacteria have functional CopA-like copper transporting proteins but no copZ-like genes, albeit while this is true of E. coli a CopZ like protein is made through a frameshifting mechanism. However, the phenotypes of copZ (or Atx1) deleted strains are subtle with CopA ATPases retaining the ability to export copper. In vitro experiments show that CopA-like ATPases can accept copper from other sources.

While it is true that copper ions are all bound in the cytosol of bacterial cells (even strains without CopZ) binding need not be solely to proteins and could involve small molecule ligands such as glutathione.

Have the authors tried to model copper-glutathione complexes in place of the metal binding region of CopZ within the space revealed in the E1 structure?

The work of Svetlana Lutzenko has shown regulatory interactions between the amino-terminal metal-binding domains of copper-transporting P1B-type ATPases and loops associated with ATP-binding and hence enzyme turnover. These interactions inhibit activity. Upon binding of copper, copper-Atx1 (or by analogy copper-CopZ) to the amino-terminal metal-binding domains the inhibitory interaction appears to be lost. How might these observations relate to the rotations of the N and A domains? Is it possible to model how association with the amino-terminal metal-binding regions (which are structurally similar to CopZ) might lock the A- and N- domains in orientations that prevent the observed rotations and hence enzyme turnover.

We thank the reviewers for the evaluation and for the helpful suggestions and comments to improve the manuscript. We have now addressed all comments and amended the manuscript as outlined below, with the changes highlighted in yellow in the main manuscript. Remarks and questions from the reviewers are shown in black. Our responses are shown in green.

Reviewer #1 (Remarks to the Author):

The study reports an interesting and significant new step in determining the structural basis of metal transport by P1B-type ATPases. The data are clear and well-illustrated; the Supplement is very helpful. The only concern that I have is that the manuscript is somewhat difficult to read, especially for those outside the P-type ATPase field. The constant references to SERCA makes it hard to get a clear picture of how the AfCopA looks on its own and how it may work. Specifically:

1) Section "A unique E1 conformation"

a) References to previous SERCA and P1B-ATPase structure in the same paragraph are confusing. It could be more helpful to the reader if the authors first show that the current conformation of AfCopA is new/different compared to the previously published P1B-ATPase structures and after that compare it to known E1 conformations of other subfamilies to establish its uniqueness.

We would like to thank the reviewer for the suggestion. However, the reason for why we structured the first paragraph as it is, is that we first aimed for identification of the overall conformation of the determined AfCopA structure. As SERCA is the most well-studied P-type ATPase family member, this is accomplished by comparing to SERCA. After having established that the determined AfCopA structure is E1, we could analyze the E2.Pi → E1 transition in detail (Figure 2). In this way, the section goes from high-level (overall domain comparison to other P-types, Fig. 1) to low-level (comparison to LpCopA, Fig. 2). We believe that both variants to structure the section are equally suitable, but decided to leave the text and order of figures unchanged. We also note that reviewers #2 and #3 state that the manuscript is well-written and easy to follow, contrasting to the opinion of reviewer #1.

b) "...unique mechanistic features of P1A- and P1B-ATPases". While it is fair to point to some structural similarities between the two subfamilies of pumps, the overall structural and functional differences between them are too large and do not support the authors suggestions about their "unique mechanistic features". Please delete or re-phrase
Thank you for your comment. We agree that there are immense structural and functional differences between P1A- and P1B-ATPases. However, we wanted to emphasize in this paragraph that we are aware of the structural similarities – as illustrated in Supplementary Fig. 7. As this manuscript is the first E1 structure of a P1B-ATPase, these similarities have not yet been observed before and are, in our opinion, worthwhile mentioning. In agreement with the suggestion of the reviewer we have rephrased the paragraph to highlight the differences of P1A- and P1B-ATPases.

2) Section: "The A-domain modulates the catalytic cycle".

a) Again, everything is intermixed. The authors may like to consider the following organization of the paragraph: "Structural comparisons to the previously determined E2.Pi structure of LpCopA disclose major rearrangements during the E2 ↔ E1 transition (Fig. 2). Most noticeably, a large A-domain rotation of approximately 100° relative to the P-domain moves the conserved (T/S)GE dephosphorylation motif from the catalytic aspartate of the P-domain, positioning it immediately in the vicinity of the end of M4, where it is stabilized by electrostatic interactions with the P-domain (Supplementary Fig. 8). The displacement of the A-domain enables a tilt of the N-domain, closing the cytosolic headpiece, priming the nucleotide binding pocket for ATP binding (Supplementary Fig. 9)" After that one can talk about similarities and differences to other pumps.

Thank you for the advice! We have now re-arranged the section following the reviewer's suggestions.

b) "... . In agreement with this notion, the arrangement of the soluble domains has highest similarity". This phrase is confusing, since the authors stated in the previous section that this new conformation is unique. Please either clarify, reconcile with the above section, or delete

Crystal structures of the AfCopA soluble P- and N-domains in the presence and absence of nucleotide have been reported previously. With this sentence we intended to point out that the arrangement of the P- and N-domain of AfCopA in our E1 structure is more similar to the previously reported structure generated in absence compared to presence of nucleotide (highest similarity = lowest RMSD). However, we do agree that the notion is rather misleading at this point and deleted the sentence following this reviewer's suggestion.

c) In SERCA, this connecting stretch orchestrates the interspace.. - What does it mean "orchestrates"? - occupies? controls the volume of the space? Please clarify

The linker determines the distance between M1 and the distal part of the A-domain. We rephrased the sentence as follows: "In SERCA, this connecting stretch determines the distance between M1 and the distal portion of the A-domain, leaving the distance essentially constant in-between the E2.Pi and E1 states (Supplementary Fig. 11).":

d) " It is possible to speculate that the postulated absence of counterions in P1B-ATPases leads to the need for alternative stimulation of the E2.Pi \leftrightarrow E1 transition, and the tighter restrains of the A/M1 and A/M3 linkers in the E1 state may serve such as purpose, powering the A-domain rotation required for reaching the E1 conformation" - Protons, while not transported, are required for copper transport by the vertebrate P1B pumps.

In vivo, P1B-ATPases function as emergency heavy metal exporters. The E1 \rightarrow E2 transition (copper export) is triggered by ATP and/or copper binding, but in this section, we are speculating about possible triggers of the E2 \rightarrow E1 transition. Especially for extremophilic organisms, it would not make sense for the ATPase to rely on a particular kind of counter-transport, as the required ion may not always be present under extreme conditions. We therefore propose that the architecture of the protein itself stimulates the E2 \rightarrow E1 transition. As protons are abundant even under extreme conditions, we agree that P1B-ATPase mediated copper transport likely relies on protons as well. This is the reason for why we started the sentence with "It is possible to speculate ".

Does the structure of a recently published frog P1B-ATPase have similarly tight A/M3 restrain?

The recently published frog P1B-ATPase (PMID: 35245129) was determined in E2 conformations only. Therefore, we currently do not know if frog ATP7B has a similarly tight A/M3 restrain in E1 or not. However, the E2.Pi structures are highly similar in this region, suggesting a maintained restrain.

If it does - then perhaps the noted structural differences between P1B and P2-type pumps may explain very different turnover rates, rather than differently used counter ion. Please consider and discuss

Yes, we do agree that the noted structural differences between P1B- and P2-ATPases, e.g. the lacking N-terminal A-domain extension and A/M1 linker, provide a reasonable explanation for the different turnover rates. We included this in the respective section of the manuscript. See also Supplementary Fig. 16.

3) Section "Copper uptake - conserved residues in the M-domain become accessible from the cytoplasmic side"

a)"Uptake" is a very specific term related to transport, typically, from the outside of the cells into the cytosol. It could be confusing to researchers outside of the P-type ATPase field. Please consider replacing with "Copper binding to transport site" o "Copper entry to the transport pathway"

b) "Unexpectedly, in the M-domain, MA-M2 and M3-M6 each form a.." - delete "Unexpectedly"

c) "Strikingly, in our structure, unlike the E2P and E2.P" - delete "strikingly"
Thank you for your comments – we replaced as suggested. We are not aware of any studies indicating that PIB-ATPases would work as two bundles (in the TM-domain), and therefore prefer to keep the "Unexpectedly".

4. Section "CopZ-mediated Cu⁺ delivery"

a) in the sentence "Noticeably, the determined..." please delete the " opposite to available E1 structures of SERCA and the Na⁺,K⁺, -ATPase 25,43 (Supplementary Fig. 14)" Given very different nature of transported ions this particular comparison to P2-pumps is somewhat meaningless and only distract the reader from the description of the Cu-entry site

We have deleted the respective sentence and associated Supplementary Fig. 14.

b) The section on CopZ-mediated transfer is highly speculative and should be significantly toned-down or deleted. There are several problems with this part. While it is convenient to dock the C-terminal part of CopZ, in cells the Cu transfer is mediated by the entire protein and therefore the entire CopZ has to be considered. More importantly, in order for Cu transfer to occur the coordinating ligands must be much closer - i.e. less than 3Å, and it is not clear that even at this distance a single Met158 would be sufficient to facilitate the transfer of Cu from the CxxC based site of the chaperone.

We agree that the entire CopZ protein should be considered. AfCopZ is a fusion of a C-terminal Atx1-like CxxC containing copper binding domain (CTD) and a cysteine-rich, redox-active N-terminal domain (NTD). However, this domain combination differentiates AfCopZ from all other members of the Atx1-like copper chaperone family (PMID: 17609202). No structure of full-length AfCopZ has been reported to date. Even though a crystal structure of the AfCopZ NTD is available (PDB ID 2H09), it is difficult to predict how the two soluble domains would be connected / interacting. We therefore decided to base our docking model on the CTD only, as this resembles the architecture of typical Atx1-like copper chaperones from other organisms (e.g. CopZ from *B. subtilis* or *E. hirae* (PDB IDs 1K0V and 1CPZ) or *H. sapiens* Atox1 (PDB ID 7DC1)) and can thus be applied to other P1B-ATPase members as well.

Nevertheless, we now added a homology model of full-length AfCopZ generated with AlphaFold to Supplementary Fig. 15, panel D. The new panel clarifies that AfCopZ has an additional NTD, and that the generated docking model of AfCopA(E1)-AfCopZ(CTD) provides sufficient space for the additional NTD to be present.

We also agree that in order for Cu transfer to occur, the coordinating ligands must be closer than illustrated in Fig. 4c. We would like to emphasize that the docking model was generated using pyDockWEB, which utilizes a rigid-body algorithm only. In this light, the observed AfCopA-CopZ distance is relatively small, and we believe that the minimum distance would be even smaller in vivo. Known copper binding sites in proteins containing sulphur based ligands (Cys/Met) are characterized by bond lengths in the range of 2.1-2.7 Å (Metal PDB: <https://metalpdb.cerm.unifi.it/>), with copper chaperones and transport protein possessing S-Cu(I) bond lengths in the typical range of 2.2-2.4 Å. In our model, the C148(CopZ) rotamer has a significantly decreased sulphur-sulphur distance to M158b of 2.6 Å, as shown below: The top panel is similar to Fig. 4c, but with a rotamer flip of C148 (CopZ). In that conformation, C148(CopZ)-M158(Af) is only 2.6 Å away and consistent with the ability of Met158b to engage in ligand exchange reactions with CopZ coordinating Cys ligands. In grey is a pseudoatom which is placed in the center of the 3 sulphur atoms CxxC(CopZ) and M158. Distance to C151 (CopZ) is 4.1 Å, but it is rather close to the other two sulphurs (of M158 & C148). The lower panel shows the pseudoatom in the middle. Thus, minor changes of local side chains and more accurate docking may well align be congruent with permitted copper passage from CopZ to CopA.

The reason for why we included the docking model in the manuscript is to illustrate that the determined E1 conformation would allow for direct docking of CopZ to the MB' platform, which is a major finding of our manuscript and in agreement with literature (PMIDs: 23184962 and 11594769). As we do agree with the reviewer that the generated docking model does not provide objective evidence for an atomic-level Cu uptake mechanism, we toned down the paragraph. For instance, we deleted the following sentence: "Interestingly, a Cu⁺ coordinated to the CxxC motif of CopZ is surrounded by D336 (located 6.2 Å away) and E205 (6.3 Å), while M158 is more distant (9.1 Å)". In addition, we replaced "direct transfer of Cu⁺ from CopZ to M158 of the here detected transient entry site appears likely." with "illustrating that direct Cu⁺ transfer from CopZ to M158 may well be possible".

Is there any example from the previous studies that "a suitable electronegative environment" on the acceptor protein stimulates release of copper from the chaperone? If such example exists, the authors should provide a reference.

We are not aware of any matching example from literature. But it was already demonstrated elsewhere that M158, E205 and D336 are required for copper transfer from CopZ to CopA *in vitro* (PMID: 23184962, section *Invariant Entrance Residues (Met, Glu, and Asp) Are Required for Chaperone-mediated Cu⁺ Access to TM-MBS*). The authors hypothesize that "M158, E205 and D336 establish a transient initial binding of Cu⁺ in

order to remove the ion from the chaperone". Furthermore, the environment is important, as alanine substitutions abolished the function, while cysteine replacements maintained the function.

In our docking model, the copper ion at the CxxC motif of CopZ is 5.8Å away from M158b, 6.1Å from D336 and 6.3Å from E205 – too far to form transient interactions with all side chains simultaneously. Nonetheless, it would be conceivable that transient interactions of D336 and E205 with copper assist in CopZ docking to AfCopA, and we therefore included such as statement in the "Transport and regulation mechanism" section.

Lastly, if E205 and D336 are involved in stabilizing interactions between the chaperone and the pump, one would expect that these residue would be less exposed and the interaction site will be altered in E2/E2P conformations, when chaperone presumably does not bind/deliver Cu to the transport site. Is this the case?

Yes, the putative interaction site formed by M158, E205 and D336 is significantly altered in the available E2P and E2.Pi conformations of LpCopA (PDB IDs 3RFU and 4BBJ). In the E2.Pi state for instance, the distance between D336 and E205 is 8.5 Å, while the residues are further apart in the determined AfCopA E1 state (11.5 Å). Also, the M158-D336 distance is 13.3 Å in the AfCopA E1, but only 5.9 Å in the LpCopA E2.Pi state.

To further illustrate the difference between the E2.Pi and E1 states around the chaperone docking site, we aligned MA-M2 of the LpCopA E2.Pi and AfCopA E1 states (included in the revised manuscript in Supplementary Fig. 15, panel C). From the illustration it is clear that there is not enough space for the chaperone to dock to the E2.Pi conformation, which is due to the decreased distance between MB' and the soluble domains.

Suppl. Figure 12 - "Indeed, in molecular dynamics (MD) simulations of the determined structure inserted in an in-silico membrane, M158, Cys380, and Cys382 coordinate a Cu⁺, following uptake from the surrounding aqueous environment" - Based on this simulation it is not clear whether Cys380 comes close enough to copper to provide coordination environment. Please clarify.

Indeed, the closest distance of Cys380(S) to Cu⁺ in the 20 ps simulation was around 4 Å, which is on the long side for a direct ion-coordinating distance. Therefore, we have now complemented the in-silico analyses with and two MD simulations with the Cu⁺ ion initially placed in the proposed ion-binding site (as in Fig. 4d). Here, the ion almost immediately loses contact with the antenna (Met158) and leaves the initial Cys interactions in favor of the more internally located Met711. Hence, the equilibrium simulations are in agreement with a transient uptake site that prepares for inward movement of the copper ion.

Reviewer #2 (Remarks to the Author):

Copper is essential to cells as a cofactor for many key enzymes, while excess copper is cytotoxic. Therefore, tight control of intracellular copper levels is crucial for cell. Copper-transporting Class IB P-type ATPases (CopA) are essential in this process by extruding Cu from the cytoplasm of cells. Like other P-type ATPases, CopA is proposed to transport substrate through a cyclic transition of E1-E1P-E2P-E2 states (Post-Albers model), with E1 and E2 states associated substrate with high and low affinity, respectively. By now, structural information on CopA is limited to structures of CopA from *Legionella pneumophila* (LpCopA) in E2P and E2.Pi states and frog ATP7B in E2.Pi state (a very recent published work: *Sci Adv.* 2022 Mar 4; 8(9):eabl5508). Because of lacking structures in other states, molecular mechanism of CopA remains unclear.

In this study, the authors report a cryo-EM structure of CopA in E1 conformation, and found dramatic domain arrangement in the structure. It's very exciting to see the central Cu binding site switch from facing out in previous E2pi state to facing cytosol in this structure. Furthermore, authors proposed the conformational change generate enough space for interaction with ion donating chaperones. Together with the functional results, the Cu transfer mechanism from the chaperone to the transmembrane core suggested by authors is with novel insights and pretty convincing.

Overall, this work is an important study, the paper is well written and the majority of the conclusions are supported by the data.

Several major issues were listed as below:

1. Considering the membrane embedded TMs is relative steady (only move in the membrane) during the substrate transport cycle, it would be helpful for readers to understand the conformational change of CopA in E1 and E2Pi by aligning their TMs. Actually, as shown in supplementary figure S11, the whole TM domain of CopA in E1 and E2 states are superimposable well, despite of some relative shifts between TMA-TM2 and TM3-6. Such alignment by TMs will present the relative movement of A/P/N domains, while fig.2e present the relative shift of A by P. In this way, comparison between Fig.2a and 2c could be improved by aligning either MA-M2 or M3-M6.

Thank you for your constructive suggestions! As suggested by the reviewer, we replaced Fig. 2c so that MA-M2 are aligned to Fig. 2a. We agree that in the new figure it is easier to follow how CPC (M4) and M711 (M6) shift relative to M158 (M1) during the E2 → E1 transition. In addition, we generated a new Supplementary Figure illustrating the soluble domain movements if aligned on M3-6 (Supplementary Fig. 8). Only slight re-arrangements of the P-domain occur relative to M3-6, and the movements of the A- and N-domains are overall similar to Fig. 2e.

2. Authors proposed a CopA-CopZ binding model using pyDockWEB prediction software, and suggest CopZ fits well in the groove between MB' and M2-4 of CopA. I wonder whether it's possible to confirm their direct binding by *in vitro* interaction assays, such as pull-down, SPR?

We believe that it would be possible to confirm direct binding of CopZ to AfCopA *in vitro*. The AfCopZ-AfCopA interaction has already been studied elsewhere (PMID: 23184962). The authors demonstrated that Cu⁺-loaded CopZ stimulates ATPase activity *in vitro*, and determine residues required for the interaction. Our docking model presented in this manuscript supports these previously reported biochemical data. The AfCopZ-AfCopA interaction is likely transient, which may render the isolation of the complex rather difficult. *In vivo*, the interaction would require copper-loaded CopZ combined with copper-free AfCopA. Nonetheless, we attempted soaking of the obtained high-resolution crystals with AfCopZ, but were not able to collect high-resolution data on these crystals (CopZ binding likely conflicts with crystal packing). Mutated forms of AfCopA may be required to form a more stable complex, e.g. M158A, as CopZ may be able to dock to AfCopA, but not to transfer Cu⁺ to the ATPase core. Future efforts will likely shed further light on the structural details of the CopZ-CopA complex.

Furthermore, it's also helpful to know whether the CopA still keep the CopZ binding platform in E2Pi state?

Please see the response to Reviewer #1, on the question starting with "Lastly, if E205 and D336 are involved ..."

Minor points

1. In the first line of Page 7, there is probably a duplicate word "site";

Yes, corrected.

2. TMs and key residues in Fig.2a are not labeled;

Yes, corrected.

3. It would be helpful to cite the new paper of ATP7B structure in Sci Adv.

Yes, citation added.

4. In Fig.2, please add the rotation markers to show the view from 2a to 2b, 2c to 2d.

Added.

Reviewer #3 (Remarks to the Author):

The manuscript from Salustros et al. presents novel high-resolution structures of the cytoplasmic-facing E1 conformation of a copper-specific P-type ATPase. The E1 state for this P-type ATPases has not previously been characterized structurally. The crystallization construct excluded the heavy metal binding domains. In addition to the structural characterization, the authors also performed functional experiments to identify residues involved in Cu⁺ binding, allowing them to propose that a number of conserved residues are involved in the Cu⁺ transport, including a pair of cysteines. Using homology modelling and protein-protein docking, the authors furthermore illustrate a potential model for delivery of Cu⁺ to the transporter from an intracellular chaperone. The manuscript is well-written and easy to follow and the accompanying figures are of high quality, and the data and the conclusions are compelling. I recommend publication with minor revisions.

I only have a couple of minor questions for clarification:

1) The MD simulations presented in Suppl. Fig. 12: As I understand the methods, the Cu⁺ ion was pulled from the intracellular side and into the transmembrane domain. Does the pulling stop at the suggested Cu⁺ site? If not, what happens at around 15 ps where the Cu⁺ seems to unbind? Is the Cu⁺ stable in the proposed site in (longer) unbiased simulations?

The reviewer has understood the methods section correctly. The ion starts entering the membrane domain following 15 ps. To address the last two questions, we performed two additional independent and longer simulations, starting with the copper in the Met/Cys/Cys entry site of AfCopA. According to these simulations, the entry site is indeed transient, and the ion approaches the expected M711 of M6 once it has left the entry site.

2) Fig 4, legend: "CopZ docking was performed with pyDockWEB using a homology model of the C-terminal part of AfCopZ (see Methods for details)" – I don't think these details are found in the method section?

We have added the respective information to the "structure determination and analysis" part of the Methods section.

Reviewer #4 (Remarks to the Author):

This research group have been pioneers in advancing our understanding of the structure-function of P1B-type metal-transporting ATPases. The actions of these proteins have wide significance for example in nutritional immunity, inherited disease, the bio-recovery of at-risk metals and enzyme metalation in industrial biotechnology. The structure of a family member in the E1 state was previously missing and this manuscript reveals unanticipated features pertinent to metal loading from donors and potentially metal-dependent regulation via rotations of the A- and N-domains associated with ATP-binding.

Some changes are needed to the text: The closing paragraph of the introduction states that bacterial CopA proteins are 'dependent' on copper donating CopZ chaperones, but this is not as suggested. Some bacteria have functional CopA-like copper transporting proteins but no copZ-like genes, albeit while this is true of *E. coli* a CopZ like protein is made through a frameshifting mechanism. However, the phenotypes of copZ (or Atx1) deleted strains are subtle with CopA ATPases retaining the ability to export copper. In vitro experiments show that CopA-like ATPases can accept copper from other sources. Thanks for the clarification. We have adapted the respective section in the introduction as follows: "Typically, cuproproteins deliver copper to P1B-ATPases in vivo, i.e., *Archaeoglobus fulgidus* CopZ to AfCopA (PMID: 23684646) or homologous Atox1 to human ATP7A/B"

While it is true that copper ions are all bound in the cytosol of bacterial cells (even strains without CopZ) binding need not be solely to proteins and could involve small molecule ligands such as glutathione.

Have the authors tried to model copper-glutathione complexes in place of the metal binding region of CopZ within the space revealed in the E1 structure?

Indeed, just as Cu can be delivered by CXXC of CopZ to CopA, Cu can likely be delivered by GSH-Cu to CopA, perhaps via a GSH-Cu(I)₄ tetranuclear cluster (PMID: 29101230). Considering that CopZ is larger than such possible GSH-Cu(I) forms, it can be expected that this is indeed possible. We note GSH sulphur atoms could have a similar mediating role as we propose here for the CXXC-motif of CopZ. We therefore included this possibility in the main text: "Alternatively, Cu⁺ may be provided to the ATPase core by the N-terminal HMBD, which is structurally homologous to CopZ, or small molecular ligands such as glutathione."

In addition, the relative size of a GSH monomer compared to the space revealed in the E1 structure is illustrated in the figure below. In the E1 conformation, the cysteine of the GSH monomer could easily get close enough to M158, E205 and D336 to allow for direct copper delivery. Likely there is also sufficient space for larger copper-glutathione complexes, thus it cannot be excluded that copper-glutathione complexes are involved in copper delivery to the ATPase core *in vivo*.

The work of Svetlana Lutzenko has shown regulatory interactions between the amino-terminal metal-binding domains of copper-transporting P1B-type ATPases and loops associated with ATP-binding and hence enzyme turnover. These interactions inhibit activity. Upon binding of copper, copper-Atx1 (or by analogy copper-CopZ) to the amino-terminal metal-binding domains the inhibitory interaction appears to be lost. How might these observations relate to the rotations of the N and A domains?

This is an interesting question. At a fundamental level, the observed structural changes suggest that the proposed inhibition may indeed take place via domain-domain interference, thereby preventing the necessary domain movements for turn-over. However, it remains difficult to deduce further molecular details of such an inhibition, such as in which state it occurs or how the interaction is achieved. It is possible to speculate that the inhibition exploits a surface exposed in the absence of copper (E2 states), but that it is missing in the absence of copper (E1 states).

Is it possible to model how association with the amino-terminal metal-binding regions (which are structurally similar to CopZ) might lock the A- and N- domains in orientations that prevent the observed rotations and hence enzyme turnover.

In the recently determined E2 structure of frog ATP7B (PMID: 35245129), two HMBDs are resolved, and one of them is positioned between the A- and P-domains. We have generated a new supplementary figure (Supplementary Fig. 16) in which we aligned the determined AfCopA E1 structure to the A- or P-domain of ATP7B and analyzed the relative position of the HMBD of ATP7B to the soluble domains of the AfCopA E1 structure. The alignment shows that due to the position of the A-domain relative to the P-domain in the E1 structure, HMBD binding to the E1 state in the same way as to ATP7B E2 is not possible. Ergo, HMBD release from the position between the A- and P-domains would be required prior to the E2 → E1 transition.

REVIEWERS' COMMENTS

Reviewer #1 (Remarks to the Author):

The authors provided thoughtful responses to my comments and suggestions. While I do not believe that the previously published (very limited) work on Cu transfer using the truncated chaperone provides iron-clad evidence for the chaperone-mediated transfer via the "platform", nor I am convinced that the 6Å distance between the Cu-coordinating ligands is "close-enough" for Cu transfer, I do agree that the detailed Cu transfer mechanism would have to be addressed by further studies and not in this paper. Overall, the authors' interpretations of their data are careful and appropriate, the quality of data and illustrations are excellent, and I have no further concerns.

Reviewer #2 (Remarks to the Author):

Authors have answered all of my questions. I suggest this manuscript to be accepted.

Reviewer #3 (Remarks to the Author):

I thank the authors for their clarifications and edits to the manuscript and congratulate them on this nice piece of work.

Figure 13 leaves me puzzled though. The interatomic distances between the Cu⁺ ion and the proposed coordinating residues seem very long. I appreciate that the ion moves/is pulled deeper into the transmembrane domain, but even the distance to M711 seems fairly long for the majority of the 100 ns simulations (panels d and f, generally > 4 Å). Is the constant interaction with M711 in c and e just a result of restraints of some kind, since it does not seem stable in the 100 ns simulations? What is the Cu⁺ ion interacting with instead in d and f? Does it drag in water molecules? Seeing this, I wonder how good the Cu⁺ parameters are, and whether they do not properly capture the Cu⁺-sulfur interaction? I am not asking for more simulations, just curious to know what surrounds the Cu⁺ and to hear the authors' thoughts about the quality of the Cu⁺ parameters (or other explanations for why the Cu⁺ does not seem to form proper interactions).

Reviewer #4 (Remarks to the Author):

The authors have made an excellent job of addressing my comments from the first round of review and there are no further issues.

We thank the reviewers for further evaluating our manuscript. We have now addressed the final remaining comments of Reviewer #3 and amended the manuscript as outlined below. Our responses are shown in red. The changes are high-lighted in yellow in the manuscript files.

Figure 13 leaves me puzzled though. The interatomic distances between the Cu⁺ ion and the proposed coordinating residues seem very long. I appreciate that the ion moves/is pulled deeper into the transmembrane domain, but even the distance to M711 seems fairly long for the majority of the 100 ns simulations (panels d and f, generally > 4 Å). Is the constant interaction with M711 in c and e just a result of restraints of some kind, since it does not seem stable in the 100 ns simulations?

The Reviewer is correct that the 2-ns equilibration simulations (Fig. S13c,e) contain successive release of restraints (stated in the last sentence in the Methods section).

What is the Cu⁺ ion interacting with instead in d and f? Does it drag in water molecules?

The Reviewer is correct that water molecules are part of the Cu⁺ interaction. Indeed, this is particularly clear in one of the trajectories where either three waters coordinate and the M711-Cu⁺ distance is >4Å or two waters coordinate and the M711-Cu⁺ distance is ~3.5Å, which corresponds to a direct interaction (compare new FigS13 panels D and E). In the second trajectory a water mediates Cu⁺ interaction also when two waters are present (compare new FigS13 panels G and H). We have now added water analyses to Fig. S13 and describe this also in the Results section p. 8.

Seeing this, I wonder how good the Cu⁺ parameters are, and whether they do not properly capture the Cu⁺-sulfur interaction? I am not asking for more simulations, just curious to know what surrounds the Cu⁺ and to hear the authors' thoughts about the quality of the Cu⁺ parameters (or other explanations for why the Cu⁺ does not seem to form proper interactions).

The Reviewer is correct that the MD force fields do not describe Cu⁺ interactions to sulfur in a realistic way, which is why we have focused on possible pathways rather than exact interaction distances.